# Integrated Metabolomics and Network Pharmacology to Reveal the Mechanisms of *Forsythia suspensa* Extract Against Respiratory Syncytial Virus

**DOI:** 10.3390/ijms26115244

**Published:** 2025-05-29

**Authors:** Haitao Du, Jie Ding, Yaxuan Du, Xinyi Zhou, Lin Wang, Xiaoyan Ding, Wen Cai, Cheng Wang, Mengru Zhang, Yi Wang, Ping Wang

**Affiliations:** 1Shandong Academy of Chinese Medicine, Jinan 250014, China; kkitdht@foxmail.com (H.D.); kate-66@163.com (X.D.); 18306390275@163.com (C.W.); 15625156296@163.com (M.Z.); wyi_1989@163.com (Y.W.); 2School of Pharmacy, Shandong University of Traditional Chinese Medicine, Jinan 250355, China; dingjiezhyy@163.com (J.D.); 15753332285@163.com (X.Z.); roysavior@163.com (L.W.); caiwen201766@163.com (W.C.); 3School of Chinese Materia Medica, Shenyang Pharmaceutical University, Shenyang 117004, China; dyx1366006479@foxmail.com

**Keywords:** *Forsythia suspensa* extract, Respiratory Syncytial Virus, network pharmacology, metabolomics, SPR-Biacore

## Abstract

To investigate the therapeutic impact of *Forsythia suspensa* extract (FS) on RSV-infected mice and explore its antiviral pharmacodynamic foundations. Methods: An integrated analytical approach, combining UPLC-Q-TOF/MS with network pharmacology, was employed to analyze and identify the chemical constituents in FS, particularly those exhibiting antiviral properties against RSV. The study integrated network pharmacology and metabolomics for further analysis, and molecular docking and interaction experiments were conducted to validate the pharmacodynamic mechanisms. Finally, an RSV pneumonia mouse model was employed to evaluate the therapeutic influence of FS, including pathological and immunohistochemistry assessments. Twenty-five components in FS were identified through UPLC-Q-TOF/MS analysis. Integrated network pharmacology data revealed 43 effective components and predicted 113 potential targets of FS for anti-RSV activity. Metabolomics analysis identified 14 metabolite biomarkers closely linked to RSV-induced metabolic disruptions involving pathways. Moreover, molecular docking and Biacore experiments provided additional confirmation that FS primarily exerts its effects through compounds such as rutin, quercetin, and kaempferol. Immunohistochemistry experiments demonstrated a significant reduction in the expression of relevant proteins following FS administration, affirming its capacity to ameliorate lung inflammation induced by RSV infection through the modulation of Toll-like receptor signaling pathways. The data presented in this study illustrate that FS exerts its anti-RSV effects by regulating the Toll-like receptor signaling pathway and the arachidonic acid metabolism pathway via rutin, quercetin, and kaempferol. Furthermore, the approach of combining network pharmacology with metabolomics proves to be an effective research strategy for investigating the bioactive constituents of medicinal plants and elucidating their pharmacological effects.

## 1. Introduction

Respiratory Syncytial Virus (RSV) belongs to the Paramyxoviridae family and is characterized by its single-stranded negative-sense RNA genome. Upon infection, it can induce fusion-related lesions in respiratory tract cells, hence the nomenclature Respiratory Syncytial Virus [1]. This virus was initially isolated and identified in 1956, and as of now, no specific antiviral drugs are available for its treatment [2]. Furthermore, RSV infections do not confer specific immunity, thereby placing patients at risk of reinfection [3,4]. It is responsible for hospitalizing approximately 36 million people each year [5]. Prior to the COVID-19 pandemic, RSV outbreaks displayed distinct seasonality, with widespread occurrences in temperate regions during the winter and early spring and peak incidences during the rainy season in tropical and subtropical areas [6,7]. However, following the emergence of the COVID-19 virus, RSV began exhibiting atypical seasonal recurrences, coupled with a decrease in population-specific antibodies against RSV [8,9]. Consequently, there is an urgent demand for the development of treatment drugs that are both low in toxicity and effective while remaining financially accessible.

*Forsythia suspensa* represents the dried fruit of the *Forsythia suspensa* (Thunb.) Vahl plant, a member of the Oleaceae family. Widely distributed in temperate regions of China, *Forsythia suspensa* has an extensive history of utilization in traditional Chinese medicine (TCM), with Shennong’s Classic of Materia Medica (Shennong Bencao Jing) characterizing it as ubiquitous. *Forsythia suspensa* extract (FS) possesses a mild fragrance and a subtle bitterness and is noted for its attributes in heat clearance, detoxification, anti-inflammatory properties, and dispersion of swelling [10]. Its primary application lies in the alleviation of heat in the upper respiratory tract and is acclaimed as a sacred remedy for treating sores. Its principal pharmacological effects encompass antiviral, anti-inflammatory, fever-reduction, and emesis alleviation [11,12]. FS exhibits potential in the treatment of various viral infections, and there is supporting evidence validating the antiviral effects of its extracts and active constituents, such as Lianhua Qingwen Capsules and Shuanghuanglian Oral Liquid, in the context of COVID-19 prevention and control [13,14,15]. Nonetheless, its efficacy and mechanisms of action remain incompletely elucidated.

Medicinal plants contain multiple active compounds with proven efficacy, which enable them to exert their pharmacological effects through various pathways and targets. This complexity often challenges the explanation of their molecular basis and mechanisms using conventional pharmacological methods. Network pharmacology has emerged as a novel research paradigm within the realm of TCM, facilitating the transition from empirical medicine to evidence-based medicine and enhancing traditional drug discovery strategies [16,17]. Research focused on drug interventions for diseases, grounded in network pharmacology, offers robust technical support for the comprehensive, multidimensional, and systematic evaluation of the antiviral mechanisms of TCM [18,19]. Presently, TCM formulations provide reliable therapies for viral pneumonia. However, the underlying treatment mechanisms remain incompletely elucidated and warrant further exploration [20]. Metabolomics presents a ‘top–down’ research strategy that reflects organism functions through the examination of final metabolic products, allowing for the discovery of disease-state differentiating biomarkers [21]. It can be progressively extended to assess the efficacy of treatments for viral pneumonia [22]. In light of these considerations, this study introduces a novel integrated approach combining metabolomics and network pharmacology to investigate the key targets and mechanisms of FS in the treatment of RSV pneumonia while validating its pharmacodynamic basis. A schematic representation of the specific methodology is illustrated in Figure 1.

## 2. Results

### 2.1. Component Analysis of FS

The constituents of FS samples were assessed utilizing UPLC-Q-TOF-MS. The total ion chromatograms (TIC) of FS samples in both positive and negative ion modes unveiled the comprehensive composition of all the ascertained compounds, as depicted in Figure 2. Following meticulous scrutiny involving database cross-referencing and an extensive review of pertinent literature, we successfully identified a total of 25 components, as detailed in Table 1.

### 2.2. FS Significantly Inhibits RSV Infection-Induced Lung Tissue Pathology

To investigate the potential therapeutic effects of FS against RSV infection, we conducted oral gavage administration of FS subsequent to RSV infection, as depicted in Figure 3A, using ribavirin as the positive control. Our findings revealed that FS demonstrated the capacity to ameliorate the lung inflammation induced by RSV infection. Histopathological evaluation of the treatment’s efficacy unveiled that RSV infection led to substantial thickening of alveolar walls, evident vascular congestion, and a pronounced presence of inflammatory infiltrates in the lung interstitium. However, post-administration of FS and ribavirin, while some inflammatory infiltrates persisted, the structural integrity of the lung tissue remained intact, and pathological changes in the mouse lungs were mitigated, as illustrated in Figure 3B.

### 2.3. Metabolomic Analysis

#### 2.3.1. PCA

The electron spray ionization (ESI) of quality control (QC) samples obtained from lung tissues, acquired in both positive and negative ion modes, revealed well-defined peaks and a comparatively even distribution under the given detection conditions; see Figure 4A. The clustering of mixed QC samples in the QC sample assessment and PCA score outcomes demonstrated robust instrument stability and dependable data quality throughout the metabolic analysis; refer to Figure 4B. These findings imply that the administration of FS significantly mitigates the metabolic disruptions provoked by RSV infection, bringing them closer to normal levels.

#### 2.3.2. Identification of Biomarkers

To discern potential metabolites contributing to discrimination, OPLS-DA models were constructed for three distinct groups: the control group (Z), the model group (M), and the group administered with FS (L), in both positive and negative ion modes as depicted in Figure 4C. During the permutation tests, R2Y and Q2 exhibited values greater than 0.5 and approached 1, signifying the model’s robust predictive accuracy and the absence of overfitting. Based on these findings, we can confidently affirm the successful establishment of the RSV pneumonia model in this study, further underscored by the notable therapeutic effects observed post-FS administration.

VIP values obtained from the OPLS-DA models were utilized to assess the impact strength and explanatory capacity of the expression patterns of diverse metabolites concerning the discrimination of sample classifications. In this study, we conducted volcano plot analyses for the two groups (see Figure 4D) using the following criteria: VIP values greater than 1, Fold Change (FC) exceeding 1.2 or falling below 0.83, and a significance level of *p*-values < 0.05. Substances situated farther from the center in the volcano plots indicate more substantial differences. As a result, we identified a total of 14 potential differential biomarkers, as illustrated in Table 2.

#### 2.3.3. Metabolic Pathway and Network Analysis

To investigate the impact of FS on the metabolic pathways in RSV-infected mice, we entered the HMDB IDs of the identified biomarkers into MetPA (https://www.metaboanalyst.ca/ (accessed on 18 May 2023)) for metabolic pathway analysis. The findings unveiled pivotal metabolic pathways influenced by RSV intervention and FS administration, which encompassed arachidonic acid metabolism, beta-alanine metabolism, glutathione metabolism, linoleic acid metabolism, and purine metabolism; see Figure 4E. Noteworthy biomarkers associated with these metabolic pathways included leukotriene C4, prostaglandin I2, spermine, uracil, inosine, hypoxanthine, and xanthosine. This analysis provides valuable insights into the intricate metabolic interactions influenced by FS and RSV in the context of these specific pathways.

### 2.4. Network Pharmacology Analysis

#### 2.4.1. Interaction Between Compounds and Targets

To further explore the mechanism of FS against RSV, a network pharmacological investigation was undertaken. A total of 43 effective compounds were compiled and deduplicated from 25 components identified through UPLC-Q-Exactive-MS (refer to Table 1), alongside 18 components sourced from TCMSP, as documented in Table 3. Subsequently, Swiss Target Prediction was employed to retrieve potential targets for each compound. The top 100 disease-related targets were integrated from various reputable sources, including CTD, GeneCards, OMIM, PubMed, GenCLip3, and our own database, resulting in a comprehensive list of 849 targets, as detailed in Appendix A. Following the alignment of these two target sets, 113 common targets were identified as potential candidates for FS against RSV. To gain further insights into the potential mechanisms, a compound–target network was meticulously constructed utilizing Cytoscape, as depicted in Figure 5A. Moreover, to delve even deeper into the potential mechanisms of FS against RSV, a PPI network involving the 113 potential targets was meticulously created using the STRING database, as represented in Figure 5B.

#### 2.4.2. Enrichment Analysis

To further elucidate the pathways implicated in the anti-RSV infection mechanism of FS, we performed a KEGG pathway enrichment analysis on the obtained DEGs. Our findings indicated that following RSV intrusion into the human body, several pathways demonstrated significant enrichment, including the Toll-like receptor signaling pathway, TNF signaling pathway, RIG-I-like receptor signaling pathway, PI3K-Akt signaling pathway, and NF-κB signaling pathway; see Figure 5C. Notably, the Toll-like receptor signaling pathway exhibited the most pronounced enrichment. Our analysis has unveiled that the regulation of the Toll-like receptor signaling pathway by FS plays a pivotal role in its anti-RSV effects and is associated with 15 potential targets, including TLR4, TLR9, AKT1, and MAPK14; refer to Figure 5D.

### 2.5. Integrated Pathway Analysis of Targets and Metabolites

MetScape can integrate metabolomics datasets to map identified metabolites to their corresponding target proteins or genes [23]. Using MetScape to visualize the upstream targets and associated metabolites of the selected differential metabolites, see Figure 6A, a total of 81 targets were obtained. Further analysis revealed that three metabolic pathways, including arachidonic acid metabolism, glycerol phospholipid metabolism, and leukotriene metabolism, are crucial metabolic pathways in which FS is involved in regulating RSV infection. Key targets and metabolites are listed in Table 4. Finally, we constructed relationships among the chemical components of FS, target proteins, pathways, and metabolites and visualized them using Cytoscape, as illustrated in Figure 6B.

### 2.6. Immunohistochemistry

TLR4 and p38 MAPK constitute pivotal components of the Toll-like receptor signaling pathway. Biacore experiments have unequivocally established their robust binding affinities with bioactive compounds, including quercetin and kaempferol derived from FS. In order to assess the in vivo impact of FS administration, we employed IHC to scrutinize alterations in the expression profiles within lung tissues, as depicted in Figure 7 Our investigation revealed that, in RSV-infected mice, the expression levels of these aforementioned targets were significantly elevated in comparison to the control group, as evidenced by the presence of numerous positively stained cells within lung tissues. Following FS administration, a noticeable degree of downregulation was observed, with the H-FS and ribavirin treatment groups exhibiting more pronounced trends.

### 2.7. Interaction Validation

To further investigate the interplay between FS and key molecular targets, we conducted molecular docking and Biacore molecular interaction experiments aimed at elucidating the crucial amino acid residues, binding energies, and affinities of the biologically active constituents pertaining to their interactions with the central targets of the Toll-like receptor signaling pathway. This investigation encompassed the evaluation of six FS components, specifically quercetin, luteolin, forsythoside A, phillyrin, hispidulin, and kaempferol, in conjunction with their respective target proteins. Notably, the assessment of binding energies served as the primary screening criterion for the docking results, with interactions having binding energies of ≤−5 kcal/mol being indicative of robust compound–target interactions, as depicted in Appendix A. The outcomes of these investigations revealed that rutin exhibited the most robust docking activity, with the most favorable docking poses in relation to multiple target genes, except for MAPK8 and TNF. It is worth noting that MAPK8 exhibited the most optimal docking performance with wogonin, while TNF demonstrated superior docking performance when interacting with quercetin, as demonstrated in Figure 8 and Table 5.

To further substantiate the efficacy of the molecular docking method, two targets, namely TLR4 and MAPK14, were randomly selected for investigation, wherein their interactions with the aforementioned six components were assessed using Biacore. The initial screening results obtained from Biacore unequivocally demonstrated the capability of all six components to engage with the target proteins; however, it is noteworthy that the binding affinities of these compounds to the target proteins exhibited significant variability. This outcome serves to underscore the dependability of the molecular docking results. Furthermore, we opted to employ inhibitors of TLR4 and MAPK14 (positive drug: resatorvid and SB 202190) as positive control references. The findings elucidated that quercetin and kaempferol displayed the most remarkable binding affinities with these targets, surpassing even the RU values of the positive controls, and both compounds exhibited concentration-dependent interactions, as illustrated in Figure 9.

## 3. Discussion

Two years following the COVID-19 pandemic, there has been a global resurgence of RSV, marked by historically elevated hospitalization rates among infants [24]. FS constitutes a significant component of traditional antiviral Chinese medicines, such as Lianhua Qingwen Capsules and Yinqiao Powder, which have gained extensive utilization in China [25,26,27]. Nonetheless, the underlying substance basis and mechanism of action of FS in antiviral treatments necessitate further elucidation. Currently, researchers are increasingly turning to metabolomics to investigate disease mechanisms and intervention strategies. Given the capacity to amass extensive data from diverse clinical and omics models, the integration of multi-omics is emerging as a pivotal facet of metabolomics research [28]. In this study, employing FS as an exemplar, our aim was to amalgamate network pharmacology with metabolomics to assess the role of plant-based medicine in combating viral infections and discern its underlying substance basis.

We identified 14 key metabolites of FS against RSV and their related pathways in lung tissues. The results showed that FS could alleviate lung inflammation induced by RSV by affecting Toll-like receptor signaling pathways and metabolic pathways such as arachidonic acid metabolism, glycerophospholipid metabolism, beta-alanine metabolism, and purine metabolism. This intervention also improved metabolic disorders in the body, leading to the normalization of the expression levels of these 14 metabolites. This strategy provides a suitable approach for validating the results of two methods and offers insights into screening the substance basis of other drugs.

Traditional Chinese herbs contain multiple active components, offering promising prospects for the prevention and treatment of complex diseases through synergistic effects. However, the selection of therapeutically effective components remains a significant challenge in current drug development. In this study, we employed network pharmacology and molecular docking techniques to screen potential active components of FS against RSV. Subsequently, we validated the molecular docking results using Biacore, which demonstrated that quercetin, kaempferol, and rutin, among others, exhibited strong interactions with multiple Toll-like receptors. The Toll-like receptor signaling pathway emerged as a prominent signaling cascade regulated by FS in its antiviral effects. Biacore, a widely adopted method for molecular interaction analysis relying on surface plasmon resonance (SPR), leverages principles of physical optics to detect changes in the dielectric constant arising from molecular binding, resulting in alterations in response values. Biacore technology allows for the repeated utilization of target proteins as receptors, making it efficient for screening and verifying various molecules [29].

Quercetin, kaempferol, and rutin are widely found flavonoids in the plant kingdom, exhibiting notable anticancer, anti-inflammatory, antioxidant, antiviral, and immunomodulatory effects. After exploring and analyzing a large number of scientific data, some studies have found that employing bibliometric analysis via VOS Viewer unveils the novel biological activities and high efficacy of compounds such as quercetin, kaempferol, rutin, apigenin, and curcumin against pathogenic viruses including HIV, COVID-19, HBV, and RSV. The intricate mechanisms underlying their antiviral effects and medicinal applications are also discussed in greater depth [30]. Among others, they investigated the interaction between hRSV NS1 and natural flavonoids such as kaempferol and myricetin by spectroscopic techniques and computer simulations. Fluorescence data suggested an affinity of approximately 105 M-1, which was directly related to the distribution coefficients of the compounds. Thermodynamic analysis indicated that hydrophobic interactions played a critical role in the formation of the NS1/flavonoid complex [31]. Quercetin has the pharmacological characteristics of low solubility and poor specificity. Some scholars obtained its derivative molecule (Q1) by acetylating quercetin (Q0) and compared their anti-RSV effects through in vitro experiments. The results show that acetylation could enhance the antiviral activity of quercetin. Penta-acetylated quercetin could interact with F protein, with lower binding energy and better stability, which can block virus adhesion [32]. In addition, quercetin could interact with RSV NS1 protein to inhibit RSV infectivity and intracellular replication [33]. Therefore, we speculate that FS may regulate the Toll-like receptor signaling pathway mainly through rutin, kaempferol, and quercetin to achieve the anti-RSV effect.

Toll-like receptors are well-defined pattern recognition receptors responsible for pathogen recognition and induction of innate immune responses. Among these receptors, TLR4 is located on the cell surface and is one of the major membrane components recognizing microbes. Once activated, it plays a crucial role in inducing the expression of genes related to inflammatory responses in innate immunity [34] by detecting the expression levels of miR-26b and TLR4 mRNA in peripheral blood monocytes from infants with bronchiolitis caused by RSV infection. It was found that infants with RSV infection had higher levels of miR-26b and lower levels of TLR4 mRNA [35]. Furthermore, in vitro simulations confirmed that RSV infection upregulated miR-26b to inhibit TLR4 signaling, indicating TLR4 as a potential target for the prevention and treatment of RSV infection. Lipid components such as POPG and PI could antagonize the activation of TLR2 and TLR4 by their respective ligands, blocking the recognition of ligands by TLRs. This antagonistic mechanism can inhibit inflammation related to RSV and IAV infections, significantly suppressing infection [36]. The p38 kinase family is a key mediator of cellular stress responses, and its pathways are associated with diseases such as dysregulation, inflammation, and immune disorders. p38α (MAPK14), when activated by numerous signals, can regulate various cellular behaviors through multiple substrates [37,38]. Some researchers used phage display library panning technology to screen RSV single-chain antibodies (GD-scFv) to develop full-length monoclonal antibodies (GD-mAb) and evaluate their potential anti-RSV effects. The results showed that antibody treatment reduced the phosphorylation levels of proteins in the TLR4/NF-κB, MAPK, and PI3K/Akt pathways [39]. This study indicated that early p38 activation dependent on TLR4 and the mediation of multiple intracellular cascades through PI3K/AKT could lead to increased NF-κB nuclear translocation and transcriptional activity, resulting in the release of proinflammatory cytokines such as IL-6 and TNF-α.

RSV stands as a special non-cellular structured organism devoid of its own metabolic capabilities. However, it will cause metabolic changes in substances in infected cells, which is beneficial to its replication, and cells will make changes to cope with the infection process [40]. By selectively influencing specific cellular metabolic pathways, we can gain a better understanding of the metabolic changes required by the virus, laying the foundation for exploring new therapeutic approaches or clinical diagnostic markers. Biomarkers such as amino acids, nucleotides, and lipids are the core of the virus protein, genome, and envelope. The replication of RSV necessitates the integration of a multitude of nucleoside components, rendering the regulation of its metabolism and the influence on the activity of the virus RNA-dependent RNA polymerase (RdRp) as an avenue to impede viral replication [41]. Lipids, as the main constituents of pulmonary surfactant, are crucial for maintaining lung function. After invading the host, RSV causes distortions in the pulmonary surfactant system, disrupting the binding of viral particles to host cell membrane receptors and inhibiting viral uptake [42]. BALB/c mice experience metabolic disturbances in various endogenous metabolites in the lungs after RSV infection, including leukotriene C4, guanosine, spermine, phosphatidylcholine, lysophosphatidic acid, and glutathione. The intervention with FS can rectify the metabolic imbalances induced by RSV in pathways such as arachidonic acid metabolism, progressively restoring normal metabolic levels.

Arachidonic acid primarily exists in the form of phospholipids in cell membranes and is released and transformed into biologically active metabolites when the membrane is stimulated. Additionally, research suggests that arachidonic acid metabolites may partially mediate the pathogenesis of RSV infection through direct tissue damage and bronchoconstriction [43]. Upon RSV invasion, it disrupts arachidonic acid metabolism, which is believed to result from changes in the expression of phospholipids and leukotrienes, leading to immune and inflammatory responses [44]. Phosphatidylcholine (PC (18:4 (6Z,9Z,12Z,15Z)/18:4 (6Z,9Z,12Z,15Z))) is a major component of biological membranes with a choline group attached to its head. Pulmonary surfactant consists of approximately 90% phospholipids and 10% proteins and possesses innate immune activity. It can prevent alveolar collapse, reduce air–liquid interface tension, and thereby control lung infections and inflammation [45]. Recent studies have shown that pulmonary surfactant PCs are related to various physiological processes, such as Toll-like receptor-mediated innate immunity after RSV infection. Stimulation of PC synthesis is crucial in controlling viral replication [46]. Given that membrane fusion governs many cellular and viral physiological processes, it is conceivable that phospholipids can bind to the RSV envelope with high affinity, thus preventing its adsorption onto epithelial cells [47]. In this study, the FS group showed significantly increased PC levels compared with the model group, and lysophosphatidic acid (LPA) levels returned to those of the normal group. This may indicate that under drug intervention, PC is involved in the regulation of immunity.

Leukotriene C4 (LTC4) serves as a potent inflammatory mediator produced within the arachidonic acid metabolic pathway, catalyzed by LTC4 synthase, which amalgamates glutathione with LTA4. LTC4 emerges through the transcellular metabolic processes of neutrophils, macrophages, mast cells, and platelets. When recruited to sites of inflammation and activated, LTC4 plays a pivotal role in promoting mucus secretion and eliciting airway responses. By establishing an in vitro model of RSV infection using human bronchial epithelial cells (16-HBECs), the expressions of LTC4 mRNA and cysteine leukotriene (CysLT) in the model were detected by fluorescence quantitative polymerase chain reaction and enzyme-linked immunosorbent assay, respectively. The findings revealed that RSV infection upregulated LTC4 mRNA expression in HBECs, resulting in elevated CysLT secretion [48]. Previous research suggested that this may lead to airway smooth muscle contraction, increased vascular permeability, and a strong correlation with the severity of asthma, the extent of airway inflammation, and respiratory rate. The LTC4 levels in the model group were significantly higher than those in the control group, aligning with clinical research. FS notably reduced LTC4 levels, suggesting its potential effects in regulating inflammatory and immune metabolic pathways.

## 4. Materials and Methods

### 4.1. Experimental Materials

#### 4.1.1. Main Instruments

UHPLC system (AB SCIEX, Framingham, MA, USA); Triple TOF 5600 mass spectrometer (AB SCIEX Corporation, Framingham, MA, USA); ACQUITY UPLC HSS T3 column (100 mm × 2.1 mm i.d., 1.8 µm; Waters, Milford, MA, USA); New Classic MF MS105DU electronic balance (Mettler Toledo, Greifensee, Switzerland); Wonbio-96c multiple-sample cryogenic grinder (Wonbio Co., Ltd., Shanghai, China); Centrifuge 5424 R and Centrifuge 5430R refrigerated centrifuges (Eppendorf, Hamburg, Germany); LNG-T88 tabletop rapid centrifugal concentrator dryer (Taicang Huada Experimental Instrument Technology Co., Ltd., Taicang, China); JXDC-20 nitrogen blow dryer (Shanghai Jingxin Industry Development Co., Ltd., Shanghai, China); SBL-10TD temperature-controlled ultrasonic cleaner (10L) (Ningbo Xinzhi Biotechnology Co., Ltd., Ningbo, China); BiacoreTM T200 (Cytiva, Marlborough, MA, USA); CM5 sensor chips (Cytiva, USA).

#### 4.1.2. Reagents

Methanol (Fisher Chemical, Fair Lawn, NJ, USA); acetonitrile (Fisher Chemical, USA); formic acid (CNW, Düsseldorf, Germany); isopropanol (Merck, Darmstadt, Germany); L-2-chlorophenylalanine (≥98%, Adamas Reagent, Riehen, Switzerland); Milli-Q ultrapure water; DMSO (AMRESCO, Solon, OH, USA); PBS-P (Cytiva, USA); HBS-EP (Cytiva, USA); Amine Coupling Kit (Cytiva, USA); isoflurane anesthesia (RWD, China); TLR4 protein (R&D Systems, Minneapolis, MN, USA); p38 α-MAPK14 protein (BPS Bioscience Inc., Santiago, Chile); resatorvid (MCE, Monmouth Junction, NJ, USA); SB 202190 (MCE, USA); DAB chromogenic reagent (Servicebio, Wuhan, China); primary antibodies (anti-p38 rabbit pAb and Anti-TLR4 Rabbit pAb, Servicebio, China); and secondary antibody G1213-100UL (Servicebio, China).

#### 4.1.3. Preparation and Component Analysis of FS Samples

The FS samples were procured from Jianlian Chinese Medicine Store (Bozhou Yonggang Herbal Pieces Co., Ltd., Bozhou, Anhui, China) and were sourced from Yuncheng, Shanxi. They were meticulously authenticated as the dried fruits of the Oleaceae plant *Forsythia suspensa* (Thunb.) Vahl, ensuring compliance with the quality standards stipulated in the Pharmacopoeia of the People’s Republic of China (2020 edition) [49]. The extraction process involved taking 50 g of FS, soaking it in ten times its volume of water (*v*/*w*) for 1 h, and then subjecting it to a 45-min reflux process performed twice. The resulting mixture was combined, filtered through gauze, and concentrated to yield a medicinal solution with a concentration of 1 g per milliliter (1 g/mL) of crude drug, which was subsequently sterilized and stored. For analytical purposes, 1 mL of the FS extract was extracted, and methanol was added to bring the total volume to 100 mL. After 30 min of ultrasonication and centrifugation at 6000 rpm for 10 min, the supernatant was collected through a 0.22 μm membrane filter.

The components were analyzed using UPLC-Q-TOF-MS with an injection volume of 5 μL. The mobile phase consisted of two phases: the aqueous phase (A) and the organic phase (B: acetonitrile), both of which contained a 0.1% formic acid solution. A gradient elution was performed at a flow rate of 400 μL/min according to the following schedule: 0–3.5 min, transitioning from 95% to 85% A; 3.5–6 min, from 85% to 70% A, with a 0.5 min hold; 6.5–12 min, from 70% to 30% A, with a 0.5 min hold; 12.5–18 min, from 30% to 0% A, with a 7 min hold; 25–26 min, from 0% to 95% A, maintained until 30 min. Mass spectrometry parameters were configured as follows: Sheath gas flow rate: 45 Arb, auxiliary gas flow rate: 15 Arb, Capillary temperature: 400 °C, Full MS resolution: 70,000, MS/MS resolution: 17,500, Collision energy: 15/30/45 in NCE mode, Spray Voltage: 4.0 kV (positive) or −3.6 kV (negative). Relevant information on the components of FS was systematically collected and organized from databases, such as CNKI, ChemSpider, SciFinder, ChemicalBook, and PubChem, to construct a chemical component database for FS. The data were then analyzed using Compound Discovery 3.3.3.200 software to identify molecular ion peaks and fragment ions, interpret secondary fragment ion mass spectra, and identify chemical components by integrating literature evidence to confirm their sources.

#### 4.1.4. RSV Virulent Strain

To evaluate the infectivity of the RSV strain, the 50% tissue culture infectious dose (TCID_50_) assay was conducted in Hep-2 cells. Log-phase Hep-2 cells were seeded in 96-well plates at a density of 1 × 10^4^ cells per well and cultured at 37 °C with 5% CO_2_ for 24 h to achieve 80–90% confluence. The RSV viral stock was serially diluted 10-fold (from 10^−1^ to 10^−^⁸) and inoculated into the wells, with six replicates per dilution and a normal control group. After 48 h of incubation under standard conditions, cytopathic effects (CPE) were observed under a microscope and recorded. Based on the number of CPE-positive wells at each dilution, the TCID_50_ value was calculated using the Reed–Muench method and expressed per 100 µL. This method has been widely validated and applied in recent RSV-related studies for virus titration and infection model establishment [50,51]. The final infectious titer of the RSV viral stock was determined to be 10^−5.73^ TCID_50_/100 µL.

### 4.2. Experimental Animals

Thirty-six female BALB/c mice, aged 2 weeks and weighing between 10 and 12 g, were procured from Beijing Weitong Lihua Experimental Animal Technology Co., Ltd., Beijing, China (Certificate: SCXK (Jing) 2016-0006). These mice were housed in the ABSL-2 laboratory of the Animal Experimental Center at Shandong University of Traditional Chinese Medicine. The housing conditions were carefully regulated, maintaining a room temperature between 22 and 26 °C, humidity levels within the range of 40–70%, and a lighting schedule of 12 h of light followed by 12 h of darkness. All experimental procedures received prior approval from the Animal Ethics Committee of Shandong University of Traditional Chinese Medicine (Approval No.: SDUTCM20201109001).

### 4.3. Experimental Animal Design

#### 4.3.1. RSV Experimental Model

Mice were adaptively reared for 3 days following a quarantine period. On the fourth day, they were randomly allocated into 6 groups (n = 6): a normal group, a model group, a positive control group (administered ribavirin), and high-, medium-, and low-dose groups of FS (referred to as H-FS, M-FS, and L-FS). The mice were anesthetized using isoflurane inhalation, and 50 μL of RSV viral fluid containing 100 TCID50 was intranasally administered to establish the infection model. The normal group received 50 μL of maintenance medium intranasally as a control. Intranasal administration was carried out once daily for 3 consecutive days.

#### 4.3.2. Dosing Regimen

The clinical standard dose of ribavirin for humans is 450 mg/day, while the standard dose range for FS in humans falls between 6 and 15 g/day. To determine equivalent doses for mice, the following calculations were employed: the ribavirin equivalent dose for mice was established at 70.24 mg/kg·d, and for FS, the high, medium, and low doses for mice were computed at 4013.86 mg/kg·d, 2006.93 mg/kg·d, and 1003.47 mg/kg·d, respectively, using the dose conversion formula for experimental animals [52]. The normal group received a matching volume of physiological saline, administered at a dosage of 10 mL/kg, once daily for a span of 5 days, with daily body weight measurements.

Calculation of Dosing for per mouse:(1)$$\mathrm{Per}\;\mathrm{mouse}’s\;\mathrm{dose}\;(\mathrm{mg}/\mathrm{kg})\;\mathrm{Db}=\mathrm{Da}\cdot \frac{Rb}{Ra}\cdot {\left(\frac{Wa}{Wb}\right)}^{\frac{1}{3}}$$

Note: In this formula, Db represents the dose per kilogram of body weight for the b group of animals to be administered, Da is the known dose per kilogram of body weight for the group of animals, Ra and Rb are size factors, and Wa and Wb are the weights of the known animal and the b animal, respectively. In this study, a represents humans, and b represents BALB/c mice.

### 4.4. Specimen Collection and Processing

Mice were euthanized at the cervical vertebra under aseptic conditions, and the chest was opened to collect lung tissues. The lung tissues were rinsed with physiological saline, excess moisture was removed with filter paper, and the degree of lesions was recorded. The left lung lobe and trachea were placed in 4% paraformaldehyde for fixation, and the remaining lung tissues were stored in a −80 °C freezer for future use.

### 4.5. Histopathology and Immunohistochemical Analysis

For histopathological examination, the left lung lobes of mice were fixed in 4% paraformaldehyde for 24 h, followed by embedding in paraffin, continuous sectioning, and subsequent deparaffinization with water. Some samples underwent sequential treatment with xylene, anhydrous ethanol, and ethanol for thorough washing. Subsequently, these sections were stained with hematoxylin and eosin (HE), dehydrated, and coverslipped. In addition, immunohistochemical (IHC) staining was also applied to evaluate the TLR4 and p38 MAPK14 levels. In brief, the sections were then subjected to scanning under a light microscope, and 5 random fields were selected for observation and photography. For the remaining samples, antigen retrieval was conducted using citric acid antigen repair buffer (pH = 6.0) via microwave heating. Endogenous peroxidase activity was quelled with 3% H_2_O_2_, followed by blocking with 3% BSA. Primary antibodies (anti-p38 rabbit pAb; anti-TLR4 rabbit pAb) were introduced and incubated at 4 °C overnight. Subsequently, secondary antibodies were applied and incubated at room temperature for 50 min. Finally, DAB staining was employed for visualization. The results were captured by light microscopy.

### 4.6. Metabolomics Study

#### 4.6.1. Sample Preparation

Fifty milligrams of lung tissue were meticulously weighed, and 400 µL of a methanol extraction solution was added to achieve a 4:1 volume ratio. This solution included 0.02 mg/mL of L-2-chlorophenylalanine, serving as the internal standard. Subsequently, the samples underwent grinding with grinding beads for a duration of 6 min, followed by the extraction of metabolites under low-temperature ultrasonication conditions at 5 °C and 40 kHz. Following this, the samples were subjected to centrifugation at 13,000× *g* and 4 °C for a duration of 15 min, resulting in the extraction of the supernatant, which was designated for UPLC-Q-TOF/MS analysis. Additionally, 20 µL of the supernatant from each individual sample was isolated and combined to form a quality control sample.

#### 4.6.2. Instrument Conditions

The characterization of metabolites was performed using UPLC-Q-TOF/MS. The mobile phase was comprised of two components: (A) 95% water + 5% acetonitrile and (B) 7.5% acetonitrile + 47.5% isopropanol + 5% water, with both solutions containing 0.1% formic acid. The gradient elution program proceeded as follows: an initial 0–0.5 min with 100% A, transitioning to 75% A and maintained for 2 min, followed by 2.5–9 min with 75% A, gradually decreasing to 0% A and held for 4 min, and finally, 13.0–13.1 min shifting from 0% A to 100% A, maintaining 100% A until 16 min. The flow rate was set at 0.40 mL/min, and a 10 μL injection volume was utilized. Subsequent to separation, mass spectrometry signals were acquired in both positive and negative ion modes through electrospray ionization (ESI) for detection and subsequent analysis.

#### 4.6.3. Multivariate Statistical Analysis

The raw data underwent processing through the Progenesis QI software (version 3.0, Nonlinear Dynamics, Waters), encompassing tasks of peak extraction, alignment, and identification, resulting in the generation of a comprehensive data matrix that incorporated retention time, peak area, mass-to-charge ratio, and identification details to facilitate subsequent data analysis. Subsequent multivariate analysis, including unsupervised Principal Component Analysis (PCA) and supervised Orthogonal Partial Least Squares Discriminant Analysis (OPLS-DA), was executed using R package (version 4.2.3) and Python (version 3.9.13). Further, mass spectrometry data were cross-referenced with metabolite databases such as HMDB (http://www.hmdb.ca/ (accessed on 23 June 2022)) and METLIN (https://metlin.scripps.edu/ (accessed on 27 June 2022)) with a mass error threshold of less than 10 parts per million (ppm). Metabolites were identified based on the matching scores derived from the secondary mass spectra. A *t*-test of *p* < 0.05 and a VIP value of >1 were used as screening criteria for differential metabolites, and the obtained differential metabolites were imported into MetPA (https://www.metaboanalyst.ca/ (accessed on 30 June 2022)) for pathway metabolism analysis.

### 4.7. Network Pharmacology and Metabolomics Integration Analysis

To construct the metabolic product–protein–pathway network and elucidate key metabolites and their associated proteins, the following series of steps were executed using Cytoscape 3.7.2. (1) Analysis and identification of components in FS extract were carried out using the Traditional Chinese Medicine Systems Pharmacology (TCMSP) platform (https://www.tcmsp-e.com/ (accessed on 5 July 2022)) in conjunction with Ultra-Performance Liquid Chromatography-Quadrupole-Exactive Mass Spectrometry (UPLC-Q-Exactive-MS). Subsequently, the targets of each component were predicted using the Swiss Target Prediction tool (http://www.swisstargetprediction.ch/ (accessed on 15 July 2022)). (2) Differentially expressed genes (DEGs) in response to RSV were obtained from the Gene Expression Omnibus (GEO) dataset GSE32138, which documented the effect of RSV infection on human airway epithelial cells [53]. To identify potential targets for RSV pneumonia, we performed keyword-based searches in the Comparative Toxicogenomics Database (CTD, https://ctdbase.org/ (accessed on 19 July 2022)), Online Mendelian Inheritance in Man (OMIM, https://omim.org/ (accessed on 25 July 2022)), GeneCards (https://www.genecards.org/ (accessed on 29 July 2022)), GenClip3 (http://cismu.net/genclip3/analysis.php (accessed on 1 August 2022)), and the National Center for Biotechnology Information (NCBI, https://www.ncbi.nlm.nih.gov (accessed on 3 August 2022)). The top 100 DEGs from the gene expression data were incorporated to establish the RSV pneumonia target database. (3) The intersection of steps (1) and (2) was performed to obtain the predicted targets of FS against RSV. Standardization of gene and protein names was accomplished using UniProtKB (http://www.uniprot.org/ (accessed on 8 August 2022)). (4) Targets were imported into String 11.0 (https://string-db.org/ (accessed on 11 August 2022)), and protein–protein interaction (PPI) network was constructed using Cytoscape 3.7.2, followed by the identification of hub genes via the CytoHubba plugin in Cytoscape. (5) Gene Ontology (GO) analysis and Kyoto Encyclopedia of Genes and Genomes (KEGG) pathway enrichment analysis for potential targets were conducted using DAVID 6.8, with a significance threshold set at *p* < 0.05. (6) The differential metabolites identified in the metabolomics analysis were imported into Cytoscape, employing the MetScape plugin to generate a compound–reaction–enzyme–gene network. Then, the crucial metabolites and proteins were recognized.

### 4.8. Molecular Docking

Molecular docking in this study primarily focused on the targets enriched within the Toll-like receptor signaling pathway, an approach grounded in network pharmacology. The 3D structures of active compounds identified from FS were acquired from PubChem and subsequently subjected to energy minimization using Chem3D 19.0.0.22. Relevant crystal structures essential for the docking studies were sourced from the RCSB Protein Data Bank and meticulously prepared through the removal of water molecules, the addition of hydrogen atoms, and the optimization of hydrogen bonds. The molecular docking process itself entailed utilizing AutoDockTools-1.5.6 to assess the small molecules’ interactions within the binding pockets of the target proteins, aiming to discover the most favorable conformation and calculating binding affinities. Visualization of the docking results for the active components and the prominent protein targets was facilitated using the PyMOL2.3.0 platform. This comprehensive methodology aligns with the conventions of academic writing within the field of medicine and ensures clarity and precision in presenting the research process.

### 4.9. Biacore Experiment

To enhance the robustness of the molecular docking findings, we investigated the affinity between active compounds and essential targets utilizing the BiacoreTM 200 biomolecular interaction analyzer (GE Healthcare, Chicago, IL, USA). Channels 2 and 4 of the CM5 chip were specifically chosen for coupling with TLR4 and MAPK14, respectively, and underwent preliminary enrichment experiments to optimize their coupling conditions. The activation of carboxyl groups on the CM5 chip surface was achieved by a balanced mixture of NHS and EDC solutions, followed by the attachment of proteins to this activated surface via amine coupling. Subsequently, a two-fold dilution of the candidate compound solution was introduced into the chip at a flow rate of 10 μL/min, allowed to bind for 60 s, and subsequently regenerated using a glycine hydrochloric acid solution at 25 °C. As a reference for calibration, Channels 1 and 3 were employed. Kinetic profiles were generated by examining the correlation between binding response values and the concentration of candidate compounds, with the determination of binding specificity relying on the quality of the curve fittings.

## 5. Conclusions

In this study, we employed network pharmacology analysis, combined with UPLC-QTOF/MS technology and data post-processing strategies for data mining, to identify 43 components in FS by comparing component information. Substances such as rutin, kaempferol, quercetin, forsythin, and astragaloside were identified as the main active compounds responsible for FS’ anti-RSV effects. Furthermore, we conducted an integrated metabolomics analysis to elucidate the changes in differential metabolites induced by RSV infection, aiming to unravel the underlying molecular basis of FS’s anti-RSV activity.

At the same time, in this study, we first developed a new integration strategy. Through dry experiment prediction combined with wet experiment verification, we explored the key targets and mechanisms of FS against RSV based on metabonomics and network pharmacology and verified them by molecular docking, molecular interaction, and IHC. Our results unequivocally support the utility of this integration strategy as an invaluable tool for unraveling the regulatory mechanisms of active components within TCM.

## Figures and Tables

**Figure 1 ijms-26-05244-f001:**
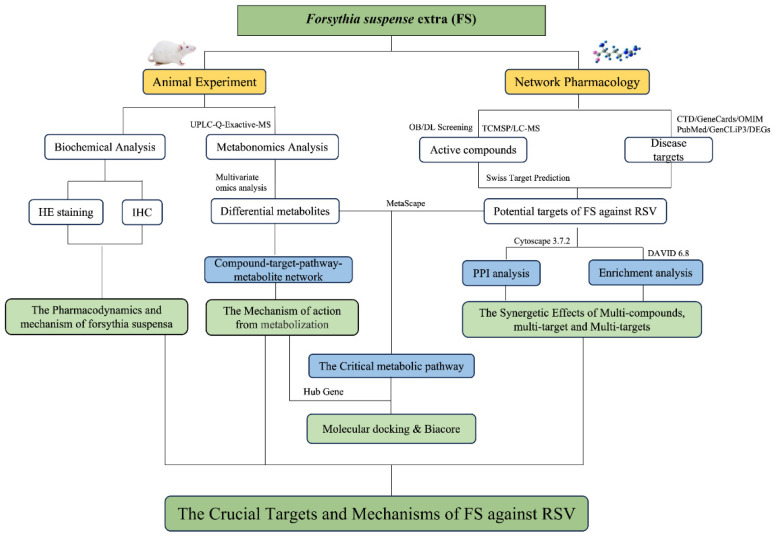
Workflow for dissecting the mechanisms of FS anti-RSV.

**Figure 2 ijms-26-05244-f002:**
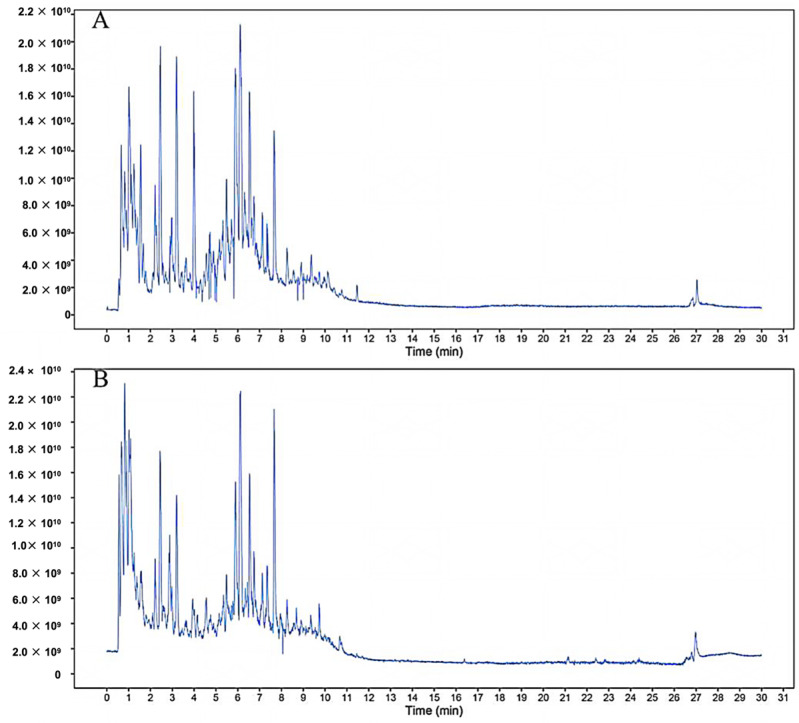
Identification of chemical components of FS. FS samples were examined by UPLC-Q-TOF-MS. Total ion chromatography (TIC) in negative (**A**) and positive (**B**) modes for FS samples is shown.

**Figure 3 ijms-26-05244-f003:**
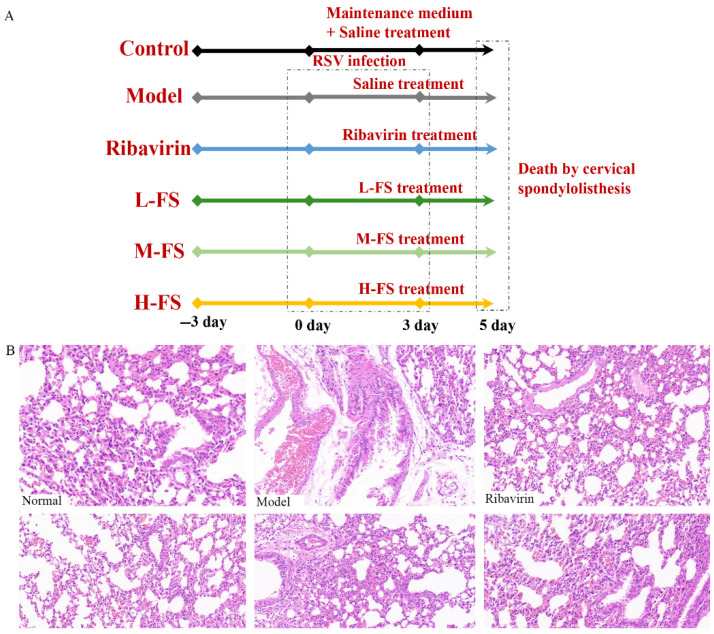
FS Exhibits Antiviral Effects Against RSV Infection. (**A**) Administration protocol; (**B**) Lung tissue pathological changes induced by RSV (200×).

**Figure 4 ijms-26-05244-f004:**
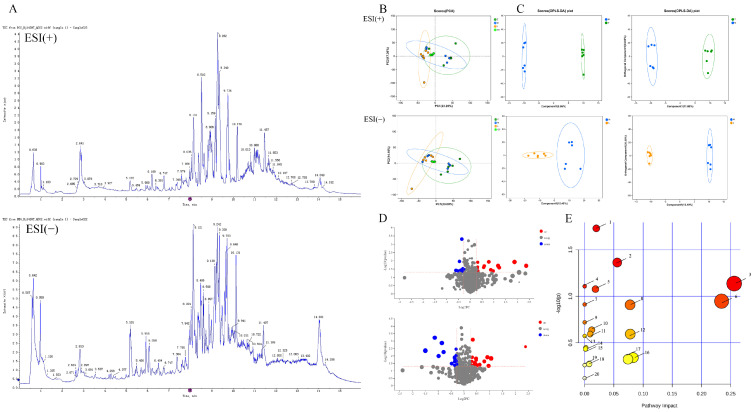
Metabolomic Analysis of Lung Tissues. (**A**) UHPLC-Q Exactive MS total ion chromatograms of FS samples in the positive and negative ion modes; (**B**) Principal component analysis (PCA) scores plot of quality control (QC) samples; (**C**) OPLS-DA scatter plot; (**D**) Volcano plot of differential metabolites in positive and negative ion modes; (**E**) Bubble plot of KEGG pathway enrichment analysis. Note: 1. Arachidonic acid metabolism; 2. beta-Alanine metabolism; 3. Glutathione metabolism; 4. Linoleic acid metabolism; 5. Purine metabolism; 6. Glycerophospholipid metabolism; 7. Ascorbate and aldarate metabolism; 8. Arginine and proline metabolism; 9. alpha-Linolenic acid metabolism; 10. Glycerolipid metabolism; 11. Starch and sucrose metabolism; 12. Pentose and glucuronate interconversions; 13. Pantothenate and CoA biosynthesis; 14. Galactose metabolism; 15. Phosphatidylinositol signaling system; 16. Amino sugar and nucleotide sugar metabolism; 17. Pyrimidine metabolism; 18. Primary bile acid biosynthesis; 19. Aminoacyl-tRNA biosynthesis; 20. Steroid hormone biosynthesis.

**Figure 5 ijms-26-05244-f005:**
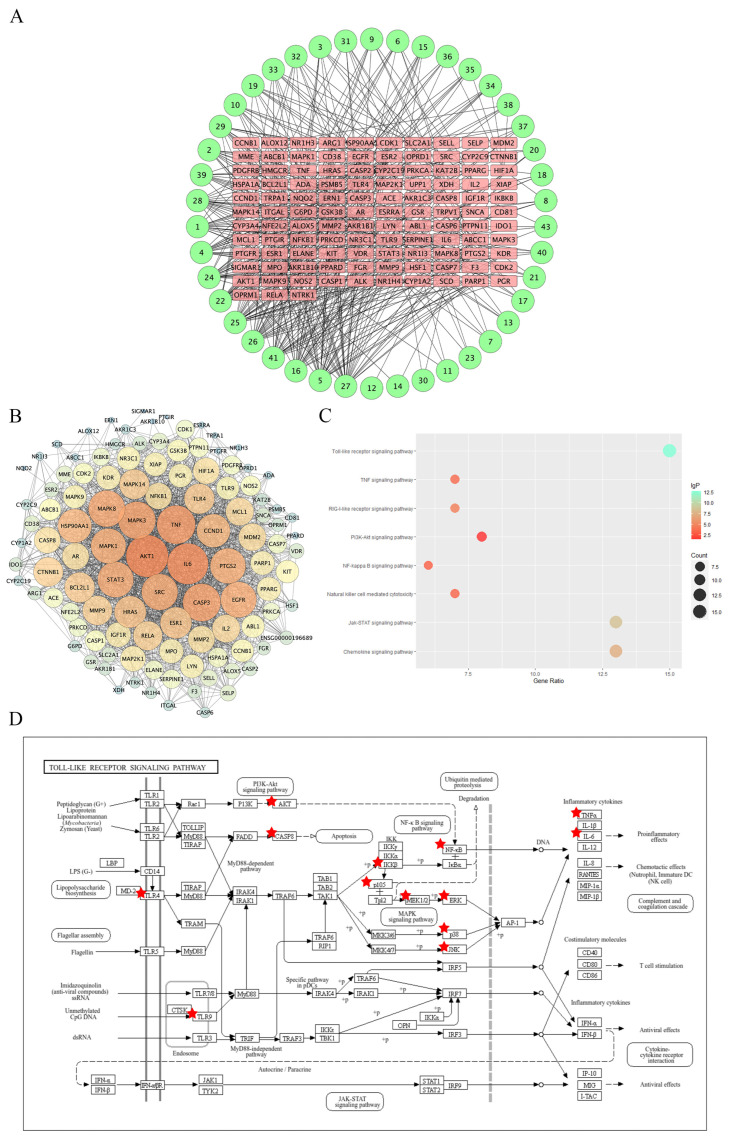
Network Pharmacology Analysis of FS against RSV. (**A**) Potential targets of FS and corresponding active compounds; (**B**) PPI network; (**C**) KEGG pathway enrichment analysis; (**D**) Enrichment results of the Toll-like receptor signaling pathway. Note: Red pentagrams indicate potential targets through which FS exerts its anti-RSV effects.

**Figure 6 ijms-26-05244-f006:**
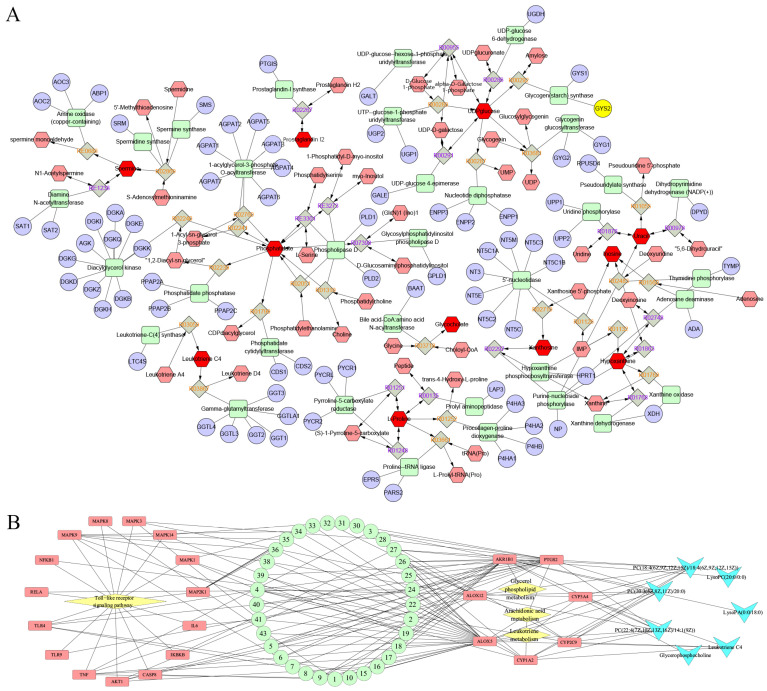
Composite Enzyme-Gene Network of Key Metabolites and Targets. (**A**) Potential targets associated with differential metabolites in FS anti-RSV based on MetScape; (**B**) Network of FS key components—targets, metabolic pathways, and metabolites. Note: Green circular nodes represent active components of FS, red rectangular nodes represent anti-RSV targets, yellow diamond nodes represent pathways, and blue V-shaped nodes represent metabolites. The network demonstrates the pharmacological basis of FS components’ anti-RSV efficacy from both the perspective of innate immunity and metabolites.

**Figure 7 ijms-26-05244-f007:**
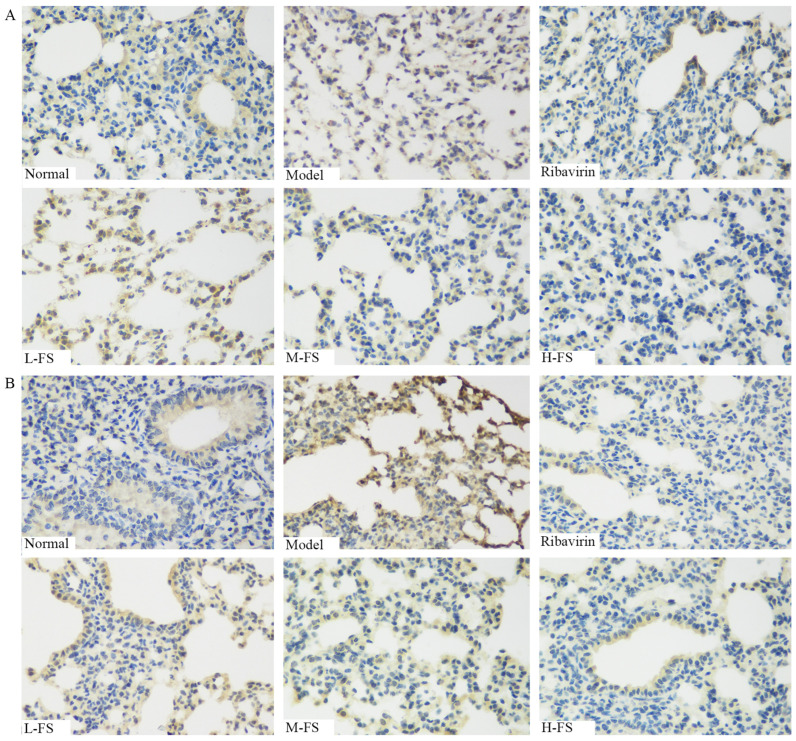
IHC images showing the protein levels of TLR4 (**A**) and p38 MAPK (**B**) in mouse lung tissues induced by RSV and inhibited by Forsythia suspensa (400× magnification).

**Figure 8 ijms-26-05244-f008:**
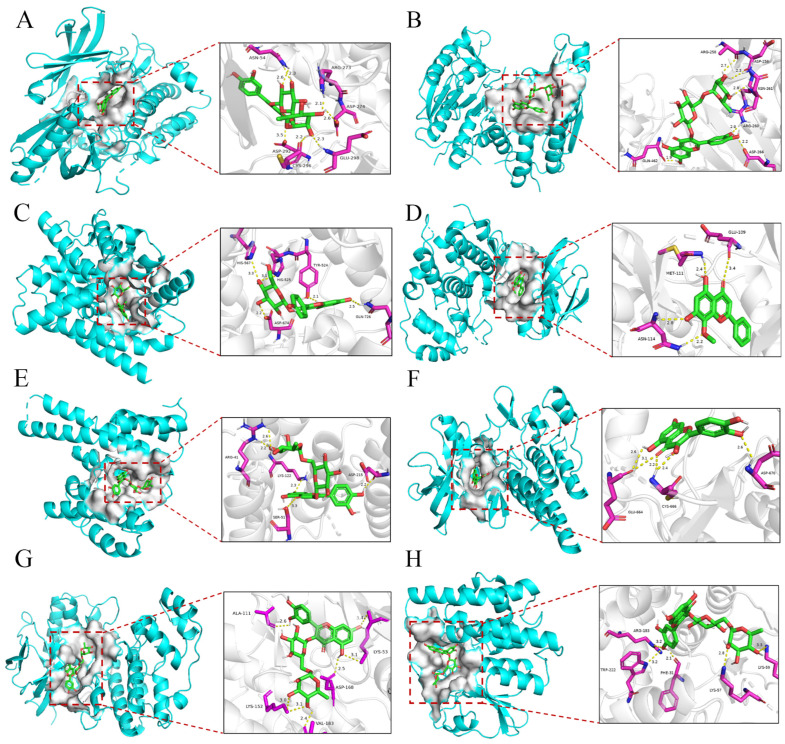
Interaction of FS pharmacodynamic components with core targets. (**A**) AKT1 with Astraglin; (**B**) CASP8 with Rutin; (**C**) IL6 with Astraglin; (**D**) MAPK8 with Wogonin; (**E**) RELA with Astraglin; (**F**) TNF with Quercetin; (**G**) p38α·MAPK14 with Rutin; (**H**) TLR4 with Rutin.

**Figure 9 ijms-26-05244-f009:**
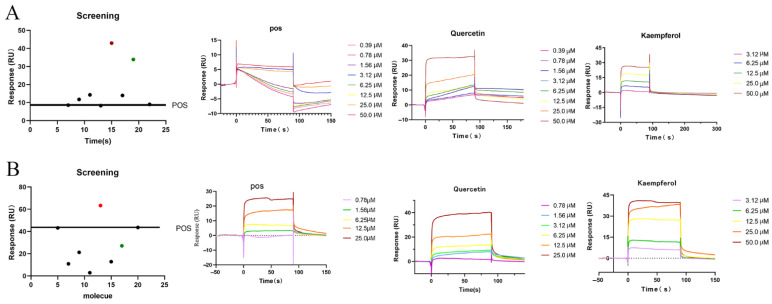
Biacore interaction assays measuring the binding of target proteins TLR4 (**A**) and p38α·MAPK14 (**B**) to components of FS. Note: The interaction results were analyzed using the kinetics and affinity method in the Kinetics Wizard template. The curves in the figure, from top to bottom, represent the response values of drugs passing over the target protein surface at decreasing concentrations.

**Table 1 ijms-26-05244-t001:** Identified Compounds by UPLC-Q-TOF-MS.

NO.	Compound	Structure	Molecular Formula	Ion Mode	Mzmed	RT/min	Peak Area
1	CITRATE	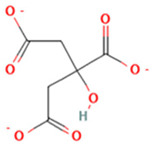	C_6_H_8_O_7_	[M − H]^−^	191.0	0.82	19,998,699,402
2	Chlorogenic acid	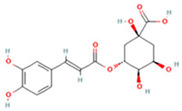	C_16_H_18_O_9_	[M − H]^−^	353.1	3.56	8,319,446,995
3	2-(3,4-dihydroxy phenyl)-5,7-dihydroxy-3-[(2S,3R,4S,5S,6R)-3,4,5-trihydroxy-6-[[(2R,3R,4R,5R,6S)-3,4,5-trihydroxy-6-methyloxan-2-yl]oxymethyl]oxan-2-yl]oxychromen-4-one	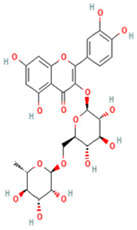	C_27_H_30_O_16_	[M + H]^+^	611.2	5.91	5,364,986,151
4	Caffeic acid	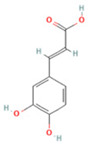	C_9_H_8_O_4_	[M−H]^−^	179.0	4.00	3,761,440,238
5	Kaempferol	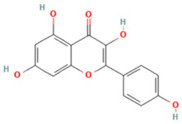	C_15_H_10_O_6_	[M + H]^+^	287.1	6.38	1,078,894,149
6	Quercetin-3-O-galactoside	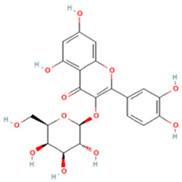	C_21_H_20_O_12_	[M − H]^−^	463.1	6.08	488,743,839.5
7	Forsythin	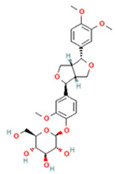	C_27_H_34_O_11_	[M + HCOO]^−^	579.2	7.73	324,696,124.3
8	Phillygenin	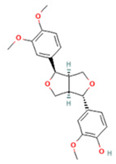	C_21_H_24_O_6_	[M + H]^+^	373.2	7.77	311,462,753.6
9	Astragalin	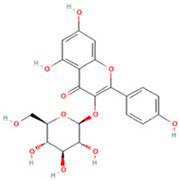	C_21_H_20_O_11_	[M + H]^+^	449.1	6.19	153,726,348
10	Cinnamic acid	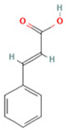	C_9_H_8_O_2_	[M − H]^−^	147.0	6.91	94,867,226.21
11	4-Methylcatechol	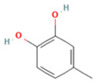	C_7_H_8_O_2_	[M − H]^−^	123.0	2.69	92,651,547.51
12	Gardenoside	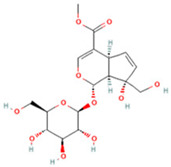	C_17_H_24_O_11_	[M − H]^−^	403.1	1.07	91,742,137.48
13	Salidroside	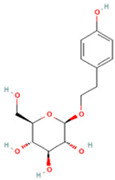	C_14_H_20_O_7_	[M − H]^−^	299.1	2.98	90,911,531.36
14	Pinoresinol 4-O-glucoside	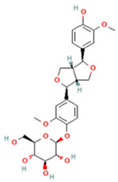	C_26_H_32_O_11_	[M − H]^−^	519.2	6.56	80,924,386.53
15	2-Hydroxycinnamic acid, predominantly trans	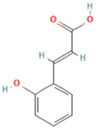	C_9_H_8_O_3_	[M + H]^+^	165.1	3.82	37,618,566.4
16	Quercetin	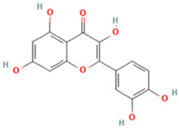	C_15_H_10_O_7_	[M − H]^−^	301.0	7.88	31,019,261.86
17	Forsythoside E	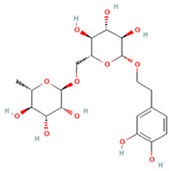	C_20_H_30_O_12_	[M + H]^+^	463.2	3.19	26,764,237.96
18	(+)-Pinoresinol	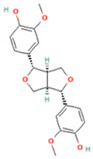	C_20_H_22_O_6_	[M − H]^−^	357.1	6.56	18,230,104.05
19	arctiin	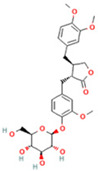	C_27_H_34_O_11_	[M − H]^−^	533.2	9.92	14,192,501.17
20	Gallic acid	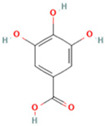	C_7_H_6_O_5_	[M − H]^−^	169.0	4.16	10,107,393.13
21	Vanillic acid	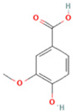	C_8_H_8_O_4_	[M + H]^+^	169.0	3.92	8,778,175.1
22	7-Hydroxycoumarin	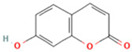	C_9_H_6_O_3_	[M − H]^−^	161.0	2.97	7,050,431.305
23	Syringic acid	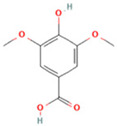	C_9_H_10_O_5_	[M + H]^+^	199.1	2.38	3,230,285.407
24	Ferulic acid	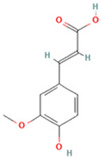	C_10_H_10_O_4_	[M + H]^+^	195.1	4.40	2,614,647.744
25	Ursolic acid	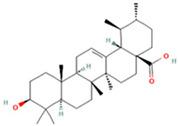	C_30_H_48_O_3_	[M + H − H_2_O]^−^	439.4	15.67	1,532,176.398

**Table 2 ijms-26-05244-t002:** Potential differential biomarkers.

NO.	Metabolite	Structure	RT	M/Z	Formula	M vs. Z	L vs. M
1	L-Proline	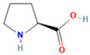	0.5312	116.0704	C_5_H_9_NO_2_	↑	↓
2	Spermine	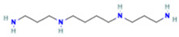	0.5507	203.2223	C_10_H_26_N_4_	↓	↑
3	Inosine	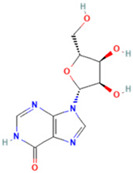	0.6451	269.0876	C_10_H_12_N_4_O_5_	↑	↓
4	Uridine diphosphate glucose	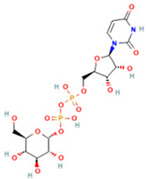	0.9191	584.0883	C_15_H_24_N_2_O_17_P_2_	↑	↓
5	Uracil	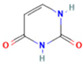	0.9776	113.0344	C_4_H_4_N_2_O_2_	↓	↑
6	Hypoxanthine	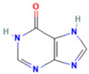	1.1476	137.0454	C_5_H_4_N_4_O	↑	↓
7	Xanthosine	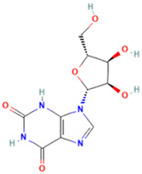	1.5070	285.0824	C_10_H_12_N_4_O_6_	↓	↑
8	Glutathionate(1-)	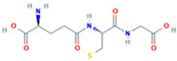	3.7882	330.0737	C_10_H_17_N_3_O_6_S	↓	↑
9	Prostaglandin I2	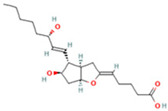	5.4683	353.232	C_20_H_32_O_5_	↓	↓
10	3a,11b,21-Trihydroxy-20-oxo-5b-pregnan-18-al	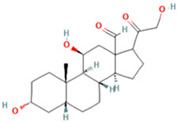	5.5731	382.2612	C_21_H_32_O_5_	↓	↑
11	Glycocholic Acid	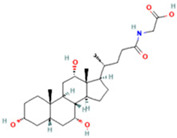	5.8629	466.3164	C_26_H_43_NO_6_	↓	↑
12	Leukotriene C4	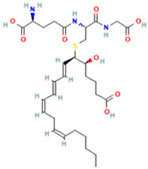	6.3286	658.3367	C_30_H_47_N_3_O_9_S	↑	↓
13	PC(18:4(6Z,9Z,12Z,15Z)/18:4(6Z,9Z,12Z,15Z))	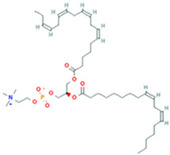	7.6699	387.7545	C_44_H_72_NO_8_P	↓	↑
14	LysoPA(0:0/18:0)	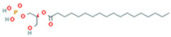	9.9456	421.2713	C_21_H_43_O_7_P	↓	↑

(↑) upregulated; (↓) downregulated.

**Table 3 ijms-26-05244-t003:** TCMSP-derived compounds.

NO.	Mol ID	Compound	MW	OB (%)	DL	HL
26	MOL000006	luteolin	286.25	36.16	0.25	15.94
27	MOL000173	wogonin	284.28	30.68	0.23	17.75
28	MOL000211	Mairin	456.78	55.38	0.78	8.87
29	MOL000358	beta-sitosterol	414.79	36.91	0.75	5.36
30	MOL000791	bicuculline	367.38	69.67	0.88	15.83
31	MOL003281	20(S)-dammar-24-ene-3β,20-diol-3-acetate	486.86	40.23	0.82	9.14
32	MOL003283	(2R,3R,4S)-4-(4-hydroxy-3-methoxy-phenyl)-7-methoxy-2,3-dimethylol-tetralin-6-ol	360.44	66.51	0.39	1.26
33	MOL003290	(3R,4R)-3,4-bis[(3,4-dimethoxy phenyl)methyl]oxolan-2-one	386.48	52.30	0.48	3.07
34	MOL003295	(+)-pinoresinol monomethyl ether	372.45	53.08	0.57	3.02
35	MOL003306	ACon1_001697	372.45	85.12	0.57	2.12
36	MOL003308	(+)-pinoresinol monomethyl ether-4-D-beta-glucoside_qt	372.45	61.20	0.57	2.90
37	MOL003315	3beta-Acetyl-20,25-epoxydammarane-24alpha-ol	502.86	33.07	0.79	7.82
38	MOL003322	Forsythinol	372.45	81.25	0.57	2.72
39	MOL003344	β-amyrin acetate	468.84	42.06	0.74	1.98
40	MOL003347	hyperforin	536.87	44.03	0.60	2.15
41	MOL003348	adhyperforin	550.90	44.03	0.61	0.84
42	MOL003365	Lactucasterol	426.75	40.99	0.85	5.53
43	MOL003370	Onjixanthone I	302.30	79.16	0.30	14.86

**Table 4 ijms-26-05244-t004:** Key target–metabolite–pathway information.

Metabolic Pathway	Differentially Expressed Gene	Potential Difference Marker
Arachidonic acid metabolism	PTGS2, ALOX5, ALOX12, CYP1A2, CYP3A4, CYP2C9	PCs
Glycerol phospholipid metabolism	AKR1B1	Glycerophosphocholine, PCs, LysoPA(0:0/18:0), LysoPC(20:0/0:0)
Leukotriene metabolism	ALOX5, CYP1A2, CYP3A4, CYP2C9	Leukotriene C4

**Table 5 ijms-26-05244-t005:** Molecular docking binding results of key targets with components.

Target	Molecule	Types of Bonds	Residues	Binding Energy (kcal/mol)
AKT1	Rutin	Hydrogen Bond	ASN-54, ARG-273, ASP-274, ASP-292, CYS-296, GLU-298	−11.6
		Hydrophobic Interaction	LEU-264, GLN-79, VAL-270, ASN-53, TRP-80	
		π-Stacking (parallel)	TRP-80	
CASP8	Rutin	Hydrogen Bond	ARG-258, ASP-259, ASN-261, ARG-260, ASP-266, GLN-462	−7.5
		Hydrophobic Interaction	TRP-420, VAL-406, ARG-258	
IL6	Rutin	Hydrogen Bond	ASP-674, GLN-726, TYR-524, HIS-525, HIS-567	−10.5
		Hydrophobic Interaction	ILE-692, PHE-696, LEU-635	
		π-Stacking (parallel)	PHE-729	
MAPK8	Wogonin	Hydrogen Bond	GLU-109, MET-111, ASN-114	−9.3
		Hydrophobic Interaction	LYS-55, VAL-40, ILE-32	
RELA	Rutin	Hydrogen Bond	LYS-122, SER-51, ASP-215, ARG-41	−8.1
		Hydrophobic Interaction	ILE-46, ILE-219, ASP-215	
TNF	Quercetin	Hydrogen Bond	GLU-664, CYS-666, ASP-670	−9.3
		Hydrophobic Interaction	ALA-614, VAL-596, LEU-588, ALA-800, LEU-785, PHE-797	
p38α·MAPK14	Rutin	Hydrogen Bond	ALA-111, LYS-53, LYS-152, VAL-183, ASP-168	−10.1
		Hydrophobic Interaction	LEU-108, VAL-30, LYS-53, TYR-182	
TLR4	Rutin	Hydrogen Bond	ARG-183, TRP-222, PHE-31, LYS-57, LYS-59	−7.8
		Hydrophobic Interaction	PHE-225, ILE-131, VAL-127	

## Data Availability

The data used in this research can be obtained upon reasonable request from the corresponding author.

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
