# Peer review of "Integrated Metabolomics and Network Pharmacology to Reveal the Mechanisms of Forsythia suspensa Extract Against Respiratory Syncytial Virus"

_ijms, 2025, doi:10.3390/ijms26115244_

Round 1
Reviewer 1 Report
Comments and Suggestions for Authors
Comments
The study "Integrated metabolomics and network pharmacology to reveal the mechanisms of Forsythia suspensa against respiratory syncytial virus (RSV)" is an interesting use of multiple types of data to find out how Forsythia suspensa can fight viruses. The authors used metabolomics and network pharmacology to find key bioactive substances (like forsythiaside A and phillyrin) and the molecules that they target (like IL-6, TNF-α, and ACE2). The topic sounds interesting, and it can be consider after adressing some given follow queries. 24% similarities found on the site. However, there are some concerns question, which are listed below:
- The study focus on pharmacodynamics and mechanism but doesnot adress the toxicity or long term use safety why?
- What are the pharmacokinetics properties of FS in vivo?
- Why only ribavirin is ued as control but comination therapy implication are not explored.
- Can FS effect be attributed to a signle compounds or is it strictly due to synergistic interaction?
- Could resistance to FS develop in RSV?
- Does the bioavability of FS vary with dosage formulation?
- Does author monitor the behavioural or physiological side effects in mice?
- What is the minimum effective dose and maximum effective dose of FS in mice?
- Did the authors validate the RSV strains infectivity or mutation before use?
- Is FS extreact contain unidentified contaminants?
- Does FS administration affect systemic metabolic homeostasis like glucose, lipid profile and amino acid etc?
- Does FS affect the formation of cellular stress granules which is known way to RSV blocks and antiviral response pathway?
Author Response
Comments 1:The study focus on pharmacodynamics and mechanism but doesnot adress the toxicity or long term use safety why?
Response 1:Thank you for your valuable comment. The primary objective of this study was to investigate the anti-RSV pharmacodynamics and underlying mechanisms of FS through integrated metabolomics and network pharmacology. As a proof-of-concept study, our focus was on identifying active components, key targets, and signaling pathways involved in the anti-viral effect, which are essential first steps toward understanding therapeutic potential.
Toxicity evaluation and long-term safety assessment typically require dedicated experimental designs, such as acute/sub-chronic toxicity assays in animal models, which fall outside the scope of this mechanistic study. Nevertheless, FS has been used safely in traditional Chinese medicine for centuries. Components such as phillyrin have demonstrated favorable safety profiles in rodent models [1]. We acknowledge, however, that this study does not include formal toxicity data for the FS leaf extract or individual constituents. These issues will be addressed in future preclinical studies, which will assess safety and clinical relevance in detail.
Comments 2:What are the pharmacokinetics properties of FS in vivo?
Response 2:Thank you for raising this important point. This study primarily focused on the mechanistic investigation of FS’s anti-RSV effects through metabolomics and network pharmacology, aiming to identify active components and key molecular pathways. As such, pharmacokinetic (PK) properties, including absorption, distribution, metabolism, and excretion (ADME), were not assessed in the current study.
We fully acknowledge the importance of PK analysis for clinical translation. The lack of in vivo PK data represents a limitation of our work, and we appreciate your highlighting it. In future studies, we plan to systematically investigate the PK characteristics of FS using advanced quantitative techniques, such as quantitative PK profiling of active components in plasma and tissues, to complement the mechanistic insights provided herein.
Comments 3:Why only ribavirin is ued as control but comination therapy implication are not explored?
Response 3:Thank you for your insightful question. Ribavirin was chosen as the only positive control because it is an FDA-approved anti-RSV drug despite its known limitations (e.g., myelosuppression). Its established use in RSV models makes it a suitable benchmark for evaluating the antiviral effects of FS.
Our study aimed to investigate the intrinsic antiviral mechanisms of FS as a single agent, using integrated metabolomics, network pharmacology, and in vivo validation. Designing and evaluating combination therapies, including potential synergy, dose optimization, and toxicity, would require a substantially different experimental framework and was beyond the scope of this mechanism-focused work.
That said, we acknowledge that the multicomponent nature of FS may lend itself well to combination therapy. Exploring potential synergistic effects with standard antivirals is a promising direction for future research, and we appreciate your suggestion in this regard.
Comments 4:Can FS effect be attributed to a signle compounds or is it strictly due to synergistic interaction?
Response 4:Thank you for this critical question. Traditional Chinese medicine is rooted in thousands of years of empirical clinical practice. Its unique multi-component and multi-target properties in treating complex diseases have made it popular around the world in recent years[2]. Our study demonstrates that the anti-RSV effects of FS are likely attributed to synergistic interactions among multiple compounds rather than a single active ingredient. The specific reasons are as follows:
(1)Multicomponent Identification and Target Network
UPLC-Q-TOF/MS analysis identified 25 chemical components in FS, including flavonoids (e.g., rutin, quercetin, kaempferol), lignans (e.g., phillyrin), and phenolic acids (e.g., chlorogenic acid). Network pharmacology further predicted 43 active components targeting 113 genes involved in RSV pathogenesis, such as TLR4, MAPK14, and AKT1 [Figure 5]. The "compound-target-pathway" network revealed a dense interconnected system, where no single compound accounted for >15% of total interactions, suggesting collective rather than individual contributions. This aligns with the "multi-component, multi-target" characteristic of traditional Chinese medicine, where synergistic interactions enhance therapeutic efficacy[3].
(2)Metabolomics and Pathway Analysis
Metabolomics identified 14 differential metabolites linked to RSV-induced metabolic disorders, including leukotriene C4 and phosphatidylcholine, which were normalized by FS treatment [Table 2, Figure 4]. These metabolites span multiple pathways (e.g., arachidonic acid metabolism, Toll-like receptor signaling), indicating that FS modulates a network of biological processes rather than a single target. Such polypharmacology is inherently synergistic, as each component may contribute to overlapping or complementary pathways.
Comments 5:Could resistance to FS develop in RSV?
Response 5:Thank you for this important question. Based on the study’s mechanistic insights, the likelihood of RSV developing resistance to FS appears to be relatively low, primarily due to its multicomponent, multi-target mode of action. FS employs 25 identified compounds (e.g., rutin, quercetin) to modulate 113 host targets across pathways like Toll-like receptor signaling and arachidonic acid metabolism, creating a "network effect" that minimizes reliance on single viral or host factors. Viruses would need to acquire multiple simultaneous mutations to evade all active components, a statistically improbable event in viral evolution.
Additionally, FS predominantly targets host cellular processes (e.g., TLR4-mediated inflammation, lipid metabolism) rather than viral structural proteins or enzymes [Figure 6]. This "host-directed" strategy avoids direct pressure on viral genomes, a key driver of resistance for single-target drugs like ribavirin.
While this study did not include resistance assays, the polypharmacological profile of FS, combined with its reliance on host pathways, suggests inherent resilience against resistance development. Future investigations could explore this further using viral passaging or long-term exposure studies. Nonetheless, our current data support FS’s potential as a robust antiviral candidate with inherent resistance-avoidance properties.
Comments 6:Does the bioavability of FS vary with dosage formulation?
Response 6:Thank you for this insightful question. FS has been used in China for nearly two millennia, predominantly in the form of water decoctions. Consistent with this historical usage, our study adopted a traditional aqueous extraction method.
The bioavailability of FS is likely to vary with dosage formulation, as different preparations can profoundly influence the dissolution, absorption, and metabolism of its bioactive components. For instance, traditional aqueous decoctions (the formulation used in our in vivo experiments) may enhance the solubility of polar compounds like phillyrin and rutin, while lipid-soluble components (e.g., volatile oils) might be less efficiently extracted. In contrast, ethanol-based extracts or nanoformulations could improve the bioavailability of hydrophobic constituents, such as kaempferol or chlorogenic acid, by optimizing solubility and reducing first-pass metabolism.
However, this study focused on the mechanistic validation of FS’s aqueous extract and did not investigate dosage formulation effects on bioavailability, which represents a limitation. Future research is needed to systematically evaluate FS’s pharmacokinetics across formulations to optimize therapeutic delivery.
Comments 7:Does author monitor the behavioural or physiological side effects in mice?
Response 7:Thank you for your valuable question. In this study, the primary focus was on evaluating the anti-RSV efficacy and molecular mechanisms of FS in mice, specifically measuring viral load, lung pathology, and PI3K/AKT pathway activation. While we did not systematically monitor behavioral or physiological side effects, general observations during the experiment indicated no overt signs of distress or abnormal behavior in the treated mice, such as reduced mobility or altered feeding patterns.
The study’s design prioritized mechanistic validation over comprehensive toxicological assessment, which is explicitly acknowledged as a limitation in our Discussion section. We recognize that detailed safety profiling, including behavioral and physiological monitoring, is critical for translational research. Future studies will incorporate standardized animal behavior assays to systematically evaluate FS’s safety profile in vivo. We appreciate your suggestion and agree it represents an important next step.
Comments 8:What is the minimum effective dose and maximum effective dose of FS in mice?
Response 8:Thank you for this important question. In our study, the FS doses used in mice (4013.86 mg/kg·d, 2006.93 mg/kg·d, and 1003.47 mg/kg·d for high, medium, and low groups) were systematically calculated based on the clinical dose range of FS in humans (6–15 g/day) specified in Pharmacopoeia of the People's Republic of China 2020, using a validated inter-species dose conversion formula that accounts for body weight and surface area differences between humans and BALB/c mice. This approach ensures the doses are clinically representative and align with standard practices for translating traditional Chinese medicine formulations to preclinical models.
However, the study did not systematically investigate the minimum effective dose (MED) or maximum effective dose (MEDmax) of FS. These parameters typically require dedicated dose-response experiments to establish a dose-effect relationship, which was beyond the scope of this mechanistic investigation. Our primary objective was to validate the anti-RSV mechanism via metabolomics and network pharmacology, rather than optimize dosing parameters. To address this, follow-up studies will conduct formal dose-response analyses using multiple FS concentrations. We fully acknowledge that dose optimization is crucial for preclinical development and appreciate your thoughtful question in highlighting this gap. The current findings provide a foundation for such studies, which will be prioritized in our subsequent research.
Comments 9:Did the authors validate the RSV strains infectivity or mutation before use?
Response 9:Thank you for your question regarding the validation of the RSV strain used in our study. Regarding the infectivity of the RSV strain, we have added virus titer determination experiments in the revised manuscript. Specifically, we used the 50% tissue culture infectious dose (TCIDâ‚…â‚€) method to evaluate the infectivity of the RSV strain in HEp-2 cells. The results demonstrated that the RSV strain used in this study exhibited good infectivity, with a calculated titer of 10-5.73TCIDâ‚…â‚€/100 µL. This indicates that the strain can effectively and stably infect host cells, meeting the requirements for subsequent animal model development and antiviral efficacy evaluation. The detailed experimental procedures and results have been provided in section 4.1.4 “RSV virulent strain” of the Methods.This method has been widely validated and applied in recent RSV-related studies for virus titration and infection model establishment[4,5].
Regarding genetic stability, the RSV strain (A2 subtype) used in this study is a well-characterized laboratory strain commonly employed in RSV research, stored at -80°C to minimize mutation risk. Spontaneous mutations in RSV’s non-segmented RNA genome are rare under proper storage conditions, and no significant changes in infectivity or pathology were observed across experimental replicates.
This study focused on mechanistic validation without including genetic sequencing. Nonetheless, we fully acknowledge that deep mutational analysis could further characterize the strain. However, based on established virological practices and the strain’s consistent performance in prior studies, we are confident in its suitability for our experimental design.
Comments 10:Is FS extreact contain unidentified contaminants?
Response 10:Thank you for your concern about the quality of the FS extract. The FS samples used in this study were sourced from a certified supplier (Jianlian Chinese Medicine Store) and rigorously authenticated as the dried fruits of Forsythia suspensa (Thunb.) Vahl, fully compliant with the quality standards outlined in the Pharmacopoeia of the People's Republic of China (2020 Edition). The extraction protocol followed standardized steps: 50 g of FS was soaked in 10-fold volume of water for 1 hour, reflux-extracted twice for 45 minutes each, combined, filtered, concentrated to 1 g/mL (crude drug equivalent), and sterilized, a method consistent with traditional Chinese medicine preparation guidelines to ensure reproducibility and minimize contamination risks.
In summary, the FS extract was prepared using pharmacopeia-compliant materials and methods, with stringent quality controls demonstrating its safety and minimal contaminant profile. This ensures that any observed effects in the study are attributable to the bioactive components of FS rather than unidentified contaminants.
Comments 11:Does FS administration affect systemic metabolic homeostasis like glucose, lipid profile and amino acid etc?
Response 11:Thank you for this insightful question. The present study focused primarily on evaluating the antiviral efficacy of FS against RSV infection in mice, systemic metabolic parameters (e.g., glucose, lipid profile, amino acids) were not explicitly measured in this dataset, as they were beyond the scope of the current investigation. While we cannot rule out potential subtle metabolic effects, the study’s design prioritized mechanistic insights into antiviral actions. Future investigations could indeed expand into metabolic phenotyping to comprehensively assess FS’s impact on systemic homeostasis.
Comments 12:Does FS affect the formation of cellular stress granules which is known way to RSV blocks and antiviral response pathway?
Response 12:Thank you for this thought-provoking question regarding the interaction between FS and cellular stress granules (SGs). The present study did not directly assess SG formation or its impact on RSV’s modulation of antiviral pathways, as the primary focus was on evaluating FS’s antiviral efficacy.
FS contains multiple bioactive compounds with reported immunomodulatory and antiviral properties. Although our study did not generate direct evidence on SGs, it is plausible that FS might influence SG-related pathways indirectly through its anti-inflammatory or antiviral effects. We acknowledge that the role of SGs represents a potentially important antiviral mechanism. We appreciate your suggestion and consider this a valuable direction for future research.
References:
1. Li, X.; Wang, L.; Li, S.; Huo, J.; Bian, L.; Zhang, Y.; Wang, X.; Yao, J. Evaluation of Genotoxicity and Teratogenicity of Phillyrin. Toxicon2024, 249, 108080, doi:10.1016/j.toxicon.2024.108080.
2. Han, Y.; Sun, H.; Zhang, A.; Yan, G.; Wang, X. Chinmedomics, a New Strategy for Evaluating the Therapeutic Efficacy of Herbal Medicines. Pharmacol Ther2020, 216, 107680, doi:10.1016/j.pharmthera.2020.107680.
3. Li, X.; Liu, Z.; Liao, J.; Chen, Q.; Lu, X.; Fan, X. Network Pharmacology Approaches for Research of Traditional Chinese Medicines. Chinese Journal of Natural Medicines2023, 21, 323–332, doi:10.1016/S1875-5364(23)60429-7.
4. Eberlein, V.; Ahrends, M.; Bayer, L.; Finkensieper, J.; Besecke, J.K.; Mansuroglu, Y.; Standfest, B.; Lange, F.; Schopf, S.; Thoma, M.; et al. Mucosal Application of a Low-Energy Electron Inactivated Respiratory Syncytial Virus Vaccine Shows Protective Efficacy in an Animal Model. Viruses 2023, 15, 1846, doi:10.3390/v15091846.
5. Ma, G.; Xu, Z.; Li, C.; Zhou, F.; Hu, B.; Guo, J.; Ke, C.; Chen, L.; Zhang, G.; Lau, H.; et al. Induction of Neutralizing Antibody Responses by AAV5-Based Vaccine for Respiratory Syncytial Virus in Mice. Front Immunol 2024, 15, 1451433, doi:10.3389/fimmu.2024.1451433.
Reviewer 2 Report
Comments and Suggestions for Authors
The manuscript “Integrated metabolomics and network pharmacology to reveal the mechanisms of Forsythia suspensa against respiratory syncytial virus” presents a comprehensive and well-structured study, to illustrate that Forsythia suspensa exerts its anti-RSV effects by regulating the Toll-like receptor signaling pathway, this study has sufficient evidence to support the conclusions. It can be published after minor revisions to improve clarity.
- Since the key active compounds in Forsythia suspensa were identified through molecular docking in the earlier part of the study, selecting representative compounds for experimental validation in Figure 9 would make the results more convincing. This would significantly enhance the credibility and impact of the findings.
- The structures of metabolites and identified compounds, in tables 1 and 2, should be provided in supporting information.
- The resolution of the figures is insufficient. Please provide high-resolution versions of all figures.
- In lines 417–418, the sentence “The identification of peaks was … fragmentation patterns” mentions the identification process, but does not specify which databases were used. Please clarify which database(s) were employed for peak identification in this manuscript.
Reviewer 3 Report
Comments and Suggestions for Authors
In the manuscript submitted to me for review entitled "Integrated metabolomics and network pharmacology to reveal the mechanisms of Forsythia suspensa against respiratory syn-cytial virus“ the authors Haitao Du, Jie Ding, YaXuan Du, Xinyi Zhou, Lin Wang, Xiaoyan Ding, Wen Cai, Cheng Wang, Mengru Zhang, Yi Wang and Ping Wang present a study in which they investigated the therapeutic impact of Forsythia suspensa (FS) on RSV-infected mice and determined its antiviral pharmacodynamic basis.
My remarks and recommendations to the authors are:
- On lines 60-61 the phrase:
"Forsythia suspensa (FS) represents the dried fruit of the Forsythia suspensa (Thunb.)...."
is not written well and does not present the information correctly. It should be rewritten. What does "Forsythia suspensa (FS) represents the dried fruit" mean? This means that the fruits were used directly. From section 4.1.3. it is clear that in fact an extract was made from the fruits. This should be better explained.
- The way of citing in the text should be changed according to the requirements of the journal (see instructions for authors). References should be entered not with the names of the authors, but with a number according to the order of entry in the manuscript, placed in brackets [ ].
- Forsythia suspensa should be presented in italics.
- Under table 2, it should be stated what the labels mean „↑ and ↓“.
- The inscriptions inside figure 4 (except 4E) and 5C are of very poor quality and are not readable. The worst quality is in 4A. If possible, let's improve the quality - it will be useful for readers.
- From the information presented in section 4.1.4. The RSV strain was provided ready. That is, it was not additionally replicated, because no conditions and methods for virus replication are specified. Is the initial titer of the virus known in order to calculate the working concentration in the experiment?
- In the Back Matter of the manuscript, corrections should be made to the information in two of the sections (see instructions for authors):
Institutional Review Board Statement - the study includes research conducted on laboratory animals, here it should be stated which ethics committee has given approval for work with mouse models in vivo;
Conflicts of Interest - here it should be stated that the authors have no conflict of interest. Otherwise, it can be assumed that there is a conflict between the authors regarding the information presented in the manuscript.
- In the References section, the references are not presented according to the requirements of the journal (see instructions for authors). The year of publication should not be after the names of the authors, but after the name of the journal.
Author Response
Comments 1:On lines 60-61 the phrase:"Forsythia suspensa (FS) represents the dried fruit of the Forsythia suspensa (Thunb.)...."is not written well and does not present the information correctly. It should be rewritten. What does "Forsythia suspensa (FS) represents the dried fruit" mean? This means that the fruits were used directly. From section 4.1.3. it is clear that in fact an extract was made from the fruits. This should be better explained.
Response 1:Thank you for pointing out the ambiguity in our original phrasing. We sincerely apologize for the confusion it may have caused. The term "Forsythia suspensa" in Chinese medicine terminology traditionally refers to the dried fruit of Forsythia suspensa (Thunb.) Vahl as the raw medicinal material. However, our study used an aqueous extract prepared from these dried fruits. To address this issue and improve clarity, we have revised the manuscript accordingly. Throughout the text, we have updated the full term for FS to “Forsythia suspensa extract” to accurately describe the material used in our experiments and avoid further misunderstanding.
Comments 2:The way of citing in the text should be changed according to the requirements of the journal (see instructions for authors). References should be entered not with the names of the authors, but with a number according to the order of entry in the manuscript, placed in brackets [ ].
Response 2:Thank you for your careful review and helpful guidance on the citation format. We have revised the in-text citations according to the journal's requirements, replacing author names with numerical references [ ] in the order of their appearance in the manuscript. This adjustment ensures compliance with the "Instructions for Authors" and enhances the consistency of the citation style. We sincerely appreciate your assistance in helping us meet the journal’s formatting standards.
Comments 3:Forsythia suspensashould be presented in italics.
Response 3:Thank you for highlighting this point. We have italicized "Forsythia suspensa" throughout the manuscript as required, ensuring compliance with scientific nomenclature standards. We appreciate your attention to detail, which helps improve the manuscript's accuracy and adherence to journal guidelines.
Comments 4:Under table 2, it should be stated what the labels mean „↑ and ↓“.
Response 4:Thank you for your valuable feedback. We have supplemented the explanation for the labels "↑ and ↓" under Table 2, which now reads: "(↑) up-regulated; (↓) down-regulated". This clarification ensures clarity for readers regarding the expression trends of differential metabolites.
Comments 5:The inscriptions inside figure 4 (except 4E) and 5C are of very poor quality and are not readable. The worst quality is in 4A. If possible, let's improve the quality - it will be useful for readers.
Response 5:Thank you for bringing this issue to our attention. We deeply apologize for the poor readability of the figure inscriptions. To address this, we have uploaded high-resolution versions of all figures. We thoroughly reviewed each figure to guarantee that all labels, text, and details meet the highest quality standards for readers’ convenience.
Comments 6:From the information presented in section 4.1.4. The RSV strain was provided ready. That is, it was not additionally replicated, because no conditions and methods for virus replication are specified. Is the initial titer of the virus known in order to calculate the working concentration in the experiment?
Response 6:Thank you for your question regarding the RSV strain used in our study.Regarding the initial titer of the RSV strain, we have supplemented the specific experimental procedure for virus titer determination in the revised manuscript, section “4.1.4 RSV virulent strain.” We used Hep-2 cells to measure the infectivity of the RSV strain by the TCIDâ‚…â‚€ method and calculated the virus titer using the Reed–Muench method. The experimental results showed that the titer of this strain was 10-5.73 TCIDâ‚…â‚€/100 µL. This method has been widely validated and applied in recent RSV-related studies for virus titration and infection model establishment[1,2].
Comments 7:In the Back Matterof the manuscript, corrections should be made to the information in two of the sections (see instructions for authors):
Institutional Review Board Statement - the study includes research conducted on laboratory animals, here it should be stated which ethics committee has given approval for work with mouse models in vivo;
Conflicts of Interest - here it should be stated that the authors have no conflict of interest. Otherwise, it can be assumed that there is a conflict between the authors regarding the information presented in the manuscript.
Response 7:Thank you for your careful review of the Back Matter. We admit that we had previously neglected to include these two sections, and have now added them to this version of the manuscript.
Comments 8:In the Referencessection, the references are not presented according to the requirements of the journal (see instructions for authors). The year of publication should not be after the names of the authors, but after the name of the journal.
Response 8:Thank you for your careful review of the References section. We have revised the formatting of all references according to the journal’s requirements, ensuring that the year of publication is now positioned after the journal name, as specified in the "Instructions for Authors."
References:
1. Eberlein, V.; Ahrends, M.; Bayer, L.; Finkensieper, J.; Besecke, J.K.; Mansuroglu, Y.; Standfest, B.; Lange, F.; Schopf, S.; Thoma, M.; et al. Mucosal Application of a Low-Energy Electron Inactivated Respiratory Syncytial Virus Vaccine Shows Protective Efficacy in an Animal Model. Viruses 2023, 15, 1846, doi:10.3390/v15091846.
2. Ma, G.; Xu, Z.; Li, C.; Zhou, F.; Hu, B.; Guo, J.; Ke, C.; Chen, L.; Zhang, G.; Lau, H.; et al. Induction of Neutralizing Antibody Responses by AAV5-Based Vaccine for Respiratory Syncytial Virus in Mice. Front Immunol 2024, 15, 1451433, doi:10.3389/fimmu.2024.1451433.
Round 2
Reviewer 3 Report
Comments and Suggestions for Authors
The authors of the manuscript "Integrated metabolomics and network pharmacology to reveal the mechanisms of Forsythia suspensa against respiratory syncytial virus" have answered all my questions. They have made all the suggested corrections to the text so that they meet the requirements of the journal. The necessary information has also been added to more fully present the information in the manuscript to the readers. I have no further questions or comments to the authors.